# Fabrication of CZTSe/CIGS Nanowire Arrays by One-Step Electrodeposition for Solar-Cell Application

**DOI:** 10.3390/ma14112778

**Published:** 2021-05-24

**Authors:** Roberto Luigi Oliveri, Bernardo Patella, Floriana Di Pisa, Alfonso Mangione, Giuseppe Aiello, Rosalinda Inguanta

**Affiliations:** Laboratorio di Chimica Fisica Applicata, Dipartimento di Ingegneria, Università di Palermo, Viale delle Scienze, Ed.6, 90128 Palermo, Italy; robertoluigi.oliveri@unipa.it (R.L.O.); bernardo.patella@unipa.it (B.P.); Floriana.dipisa@unipa.it (F.D.P.); alfonso.mangione@unipa.it (A.M.); giuseppe.aiello03@unipa.it (G.A.)

**Keywords:** nanowires solar cells, CZTSe solar cell, template electrodeposition, nanostructures

## Abstract

The paper reports some preliminary results concerning the manufacturing process of CuZnSnSe (CZTSe) and CuInGaSe (CIGS) nanowire arrays obtained by one-step electrodeposition for p-n junction fabrication. CZTSe nanowires were obtained through electrodeposition in a polycarbonate membrane by applying a rectangular pulsed current, while their morphology was optimized by appropriately setting the potential and the electrolyte composition. The electrochemical parameters, including pH and composition of the solution, were optimized to obtain a mechanically stable array of nanowires. The samples were characterized by scanning electron microscopy, Raman spectroscopy, and energy-dispersion spectroscopy. The nanostructures obtained showed a cylindrical shape with an average diameter of about 230 nm and a length of about 3 µm, and were interconnected due to the morphology of the polycarbonate membrane. To create the p-n junctions, first a thin film of CZTSe was electrodeposited to avoid direct contact between the ZnS and Mo. Subsequently, an annealing process was carried out at 500 °C in a S atmosphere for 40 min. The ZnS was obtained by chemical bath deposition at 95 °C for 90 min. Finally, to complete the cell, ZnO and ZnO:Al layers were deposited by magnetron-sputtering.

## 1. Introduction

The conversion of light into electricity through photovoltaic panels is a widespread practice for producing clean energy from a renewable source. In the last decade, there has been a strong reduction in the costs of photovoltaic panels as the result of constant research efforts toward the development of new materials and new structures to improve efficiency conversion. However, with the technologies currently employed, a well-known critical element of the supply chain is represented by raw material shortages. For this reason, significant research efforts are devoted to the introduction of substantial changes in the manufacturing process. In such regard, nanowire technology is considered a disruptive innovation with the potential of bringing savings in the consumption of raw materials, while also improving the overall efficiency of the production process. In such regard, semiconductors with appropriate optical and spectral characteristics are currently being investigated as a possible solution, due to their high availability and cost-effectiveness. Silicon (bandgap 1.1 eV) is indeed one of the most abundant semiconductor materials [1], widely used in microelectronic devices. For such reasons, although several alternative materials can be considered, silicon wafers currently constitute 90% of photovoltaic cell production, despite their low light absorption.

In recent years, many efforts have been made in order to develop new and alternative materials to silicon monocrystalline such as GaAs, CdTe, Cu In_x_ Ga_(1−x)_ Se_2_, Cu_2_ZnSn(S,Se)_4_, α-Si:H, and perovskite, with the achievement of interesting results as given in Figure 1 [2], which reports the comparison of the efficiency of such materials in single-junction solar cells compared to Si monocrystalline cells.

In particular, chalcogenide alloys of copper indium gallium selenium (CIGS) have proven to be very interesting due to a simple, low-cost deposition process and improved conversion efficiency. A high conversion efficiency in the 20% range has been demonstrated with micron-thick layers. [3]. However, due to the use of expensive and not easily available materials such as In and Ga, CIGS films have not yet been used in large-scale solar cells. In recent years, another important class of thin-film solar cells consisting of Cu_2_ZnSn(S,Se)_4_ (CZTS/Se) has been developed. This material is already being considered as a replacement for conventional absorber layers based on CuIn_x_Ga_1−x_SSe_2_ and CdTe. In fact, this substitution is required, owing to scarceness and high cost of In and Ga, and to the restrictions on handling Cd [4,5,6]. CZTS/Se materials have a significant advantage of containing earth-abundant chemical elements.

Furthermore, new designs for high-efficiency, low-cost solar cells have been studied and developed using nanotechnologies such as nanocomposite materials and nanoscale photonic layers [7]. The use of nanostructures in photovoltaic devices has thus emerged as a viable solution to overcome existing limits on efficiency. Due to the features of their three-dimensional geometrical and morphological structure, solar cells based on nanowire (NW) arrays are potentially cost-effective and efficient solar-energy-harvesting devices. Most of the current research, however, has focused on the fabrication of silicon NWs (SiNWs), while the synthesis of CZTS/CIGS nanowires [8] has only been addressed in a few works. Several methods are currently used to fabricate CZTS absorber layers, such as atomic beam sputtering [9], e-beam evaporation [10], thermal evaporation [11], magnetron sputtering [12,13,14,15], electrodeposition [16,17,18], spray deposition [19,20], pulsed laser deposition [21], and sol-gel [8,9,10,11,12]. Among these technologies, the electrodeposition method [22,23] has several advantages, since it is a simple and low-cost process that is performed at room temperature, does not require a vacuum, and ensures a large deposition area. In particular, the production of NW arrays by means of a two-step electroplating method into an anodic aluminum oxide template has been recently demonstrated [24], while CZTS deposition by the single-step electroplating method [23] had some difficulties related to the control of the composition of CZTS/CZTSe.

In this work, we discuss a process aimed at obtaining CZTS/CIGS NWs by single-step electrodeposition in a polycarbonate template, which allows the fabrication of a solar cell with an Al:ZnO/ZnS/NWs/Mo structure derived from the CZTS/CIGS-synthesized nanowires. The research has been focused on the structural, morphological, and compositional characteristics of Al:ZnO/ZnS/NWs/Mo nanowires.

## 2. Materials and Methods

The NW fabrication was carried out according to a referenced process [25,26], employing a Whatman polycarbonate (PC) membrane (Cytiva, Marlborough, MA, USA) as a template. Before the electrodeposition of the nanostructures, a thin Mo film was sputtered on one side of the membrane to make it electrically conductive, thus obtaining the back-contact surface. Subsequently, a nickel film was electrodeposited on the Mo film to act as a current collector and mechanical support. The Mo film was deposited by DC-magnetron sputtering at room temperature using a Mo target (99.95% pure), for 2 min at a pressure of 3 mTorr and a power of 60 W.

The polycarbonate membrane, used as a template for the fabrication of the NWs, had pores with a nominal diameter of 200 nm, an average thickness of about 15 μm, a pore density of 10^11^ pores/m^2^, and a porosity of 15–20%. These membranes were produced from polycarbonate films using the track-etch method, and they are typically used in micro-nanofiltration processes. The membrane morphology was analyzed by scanning electron microscopy (SEM), as reported in Figure 2. The cross-section image shows the typical interconnections of this type of template.

The polycarbonate membranes had a uniform porous capillary structure with a high distribution of interconnected pores, and their use for NW fabrication offered two main advantages:The membrane is easy to remove by chemical dissolution in dichloromethane with negligible damages to the nanostructures;The polycarbonate and the dichloromethane can be recovered, after the membrane dissolution, using a simple batch distillation [25].

The nanometric dimensions of the membrane pores allowed the electrodeposition of CZTS nanowires with a morphology that precisely replicated the features of the template.

The mechanical stability of the nanostructures was enhanced through the deposition of a nickel film by means of the potentiostatic method at E = −1.25 V (SCE) using the Watt bath (based on nickel sulphate, boric acid, and nickel chloride) as solution. The deposition was performed at room temperature for 2 h; afterward, the collector was thoroughly washed with millipore water and left to air-dry.

A three-electrode system was then produced by connecting a working electrode (WE, the PC membrane covered with Mo), a counter electrode (platinum mesh), and a saturated calomel electrode (SCE) as reference electrode, with a standard potential of 0.2412 V (NHE). The potentiostat used for electrodeposition was the PAR Potentiostat/Galvanostat (mod. PARSTAT 2273, Princeton Applied Research, Oak Ridge, TN, USA).

The nanostructures were grown by means of a pulsed current from 0 to −1.53 mA/cm^2^, which was appropriately shaped by tuning three fundamental parameters: the deposition current density, the *t_on_* (time at −1.53 mA/cm^2^), and the *t_off_* (time at 0 mA/cm^2^). The value of deposition current density was chosen to ensure the codeposition of the four species while limiting the formation of hydrogen. The *t_on_* was optimized to ensure a good amount of deposit and a low accumulation of gas inside the pores, while the *t_off_* was regulated to ensure both the gas outflow and the mass transport from the solution in order to replace the initial composition inside the pores. Electrodeposition was performed for 1 h, at room temperature, in a three-electrode cell assembled with the Ni collector as working electrode, SCE as reference, and a Pt mesh as a counter electrode. Before the electrodeposition, the cell was deaerated with nitrogen for 30 min. The nitrogen flow was maintained throughout the electrodeposition process.

To obtain CZTSe-based NWs, an electrolytic bath made of CuSO_4_, Zn(SO_4_), and SnCl_4_, as metal sources, H_2_SeO_3_ as a selenium source, and lactic acid as a ligand was used. In addition, Na_2_SO_4_ was used to improve the conductivity of the solution and to promote the growth of nanowires. In order to optimize the composition of the NWs, solutions with different concentrations of copper and different pHs were used.

For CIGSe NWs, a deposition bath consisting of CuSO_4_, In_2_(SO_4_)_3_, and Ga_2_(SO_4_)_3_ as metal sources, H_2_SeO_3_ as a selenium source, lactic acid as a ligand, and Na_2_SO_4_ as an electrolyte was used. In addition, the pH was maintained at 2.7 and solutions with different concentrations of gallium were used to optimize the composition of the NWs.

After electrodeposition, the polycarbonate was dissolved in pure CH_2_Cl_2_. This was a very challenging operation, because the deposited NWs could easily collapse and a polycarbonate coating could remain. To avoid these issues, the membrane dissolution was carried out in multiple iterations until it was completely removed.

The nanostructured junction was then obtained through the deposition of a ZnS buffer layer on the absorber, which was performed after depositing a CZTSe thin film between the nanowires in order to avoid the short circuit between the ZnS and the Mo back-contact. Additionally, a pretreatment with a 0.1 M solution of dodecylbenzenesulfonic acid (SBDS) was performed to avoid the localization of film deposition on the heads of the nanostructures (due to the low wettability of the nanostructures by the electrolyte), and thus to favor the formation of a uniform coating of the entire area of the nanostructured electrode without occluding the interstices between the NWs.

The CZTSe film was obtained by potentiostatic electrodeposition, applying a potential of −0.65 V (SCE) for 15 min at room temperature in a three-electrode cell, and using the same electrochemical bath used to obtain the nanostructures. Before starting the electrochemical deposition, the cell was degassed with nitrogen for 15 min, and the deaerated condition was maintained during the entire electrodeposition phase.

### 2.1. CBD-ZnS(O,OH) Buffer Layer Deposition

Although the CdS thin film synthesized via chemical bath deposition (CBD) is typically used as a buffer layer in CIGS/CZTSe solar cells [27,28,29], it presents some critical flaws related to the high toxicity of Cd and to the optical loss at short wavelengths due to the low bandgap (2.4 eV). Considering the mass production of CIGS solar cells, the use of CdS constitutes a critical weakness for their environmental impact. Recently, CBD of ZnS thin film has been adapted as buffer layer to take advantage of its properties such as good transparency, large bandgap (3.8 eV), low toxicity, and cost-effectiveness. These properties, combined with the efficiency of photovoltaic cells with CBD-ZnS, which is comparable to CBD-CdS [30,31,32], have recently given rise to scientific interest in this material.

The deposition process of such of CBD-ZnS(O,OH) started with a preliminary annealing of the nanostructures for 40 min at 500 °C in an inert environment. A ZnS(O,OH) layer with a thickness of 50 nm was then deposited onto the nanostructures by chemical bath deposition (CBD) using a buffer solution at pH 10.7, containing Na-citrate and triethanolamine (TEA) as complexing agents (required to prevent precipitation of Zn(OH)_2_), and Zn-acetate hydrate and thiourea as sources of zinc and sulfur, respectively. The CBD solution was then heated to 50 °C, then the stirring was stopped and the NWs were immersed. The solution was further heated to 95 °C, and the samples were maintained at this temperature for 90 min. Subsequently, the samples were rinsed with a hot solution (to avoid thermal shock) of ammonia and millipore water (1:10 by volume), and finally washed with hot millipore water [33].

### 2.2. ZnO/AZO Deposition

The ZnO/AZO deposition process was performed through the deposition of a very thin layer of intrinsic zinc oxide (ZnO) on top of a ZnS buffer layer. Undoped highly resistive ZnO is commonly deposited on the transparent conducting window layer in order to make the pn-junction less sensitive to shunts and material fluctuations of the absorber, and to protect the pn-junction from sputter damage due to the high energy of the plasma ions [1,29,30]. The ZnO film was deposited by RF magnetron-sputtering at room temperature using a ZnO target (99.999% purity, 2M, Cormano, Italy). The sputtering chamber first was depressurized to 10^−5^ mTorr using a turbomolecular pump, then before starting the ZnO sputtering process on the samples, a pre-sputtering phase was carried out for 5 min with a gas pressure of 5 mTorr and power of 120 W to clean the target. Afterward, the deposition of the ZnO was carried out for 60 min with a pressure of 5 mTorr and power of 150 W. An aluminum-doped zinc oxide (AZO) film was then deposited to act as a transparent electrode for the front contact. The AZO film was deposited via RF magnetron-sputtering at room temperature using a target of AZO (99.99% pure) with an Al_2_O_3_ content of 2% in weight. Subsequently, in order to evaluate the influence of the deposition parameters on AZO properties, the AZO film was first deposited on a silicon and glass substrate. The sputtering chamber was maintained at a vacuum level of 10^−5^ mTorr. In this case, a pre-sputtering phase also was carried out to clean the target by means of a 45 W RF generator. The deposition process was then performed with different parameters (power, gas pressure, deposition time, rotating or fixed sample holder) to evaluate their influence on the characteristics of the film.

After each fabrication step, a morphological analysis was carried out by scanning electron microscopy (SEM), using a FEG-ESEM FEI (mod. QUANTA 200, FEI, Hillsborough, OR, USA) equipped with an energy-dispersive spectroscopy (EDS) detector, to verify the composition of the NWs. The samples were also characterized by Raman spectroscopy. Raman spectra were obtained at room temperature using a Renishaw (inVia Raman 158 Microscope, Renishaw, UK) spectrometer equipped with a microprobe (50×) and a CCD detector. The excitation was provided by the 532 nm line of a Nd:YAG laser. The power of the incident beam on the sample was 5 mW, and the width of the analyzed spot for each sample was about 2 µm. Acquisition time was adjusted according to the intensity of the Raman scattering. The wave-number values had a 1 cm^−1^ accuracy. For each experiment, Raman spectra were recorded at several points of the sample to ascertain its homogeneity. Raman peaks were identified by comparison with the RHUFF database [34].

## 3. Results and Discussion

### 3.1. CZTS Nanowires

The nanostructures were grown using a nickel current collector obtained by electrodeposition on the Mo film previously deposited through the sputtering process. Nickel has an excellent resistance to oxidation, but unfortunately it has the disadvantage of a low overvoltage for the hydrogen evolution reaction (indeed, it is used as catalyst). This aspect is important because the deposition of both CZTSe and CIGS must be conducted at low pH values (from 2.26 to 3), and in favorable thermodynamic conditions for the reduction of hydrogen. This concurrent reaction has significant effects on the morphology of the nanostructures formed within confined systems such as polycarbonate membrane pores. In fact, the evolution of hydrogen inside the template pores, and its progressive accumulation, favors the formation of nanotubes rather than NWs. However, such effect needs to be avoided, because nanotubes have a lower mechanically stability compared to NWs. A key parameter to control the hydrogen reduction and thus the morphology of CZTSe nanostructures is the pH of solution. Therefore, electrochemical deposition was carried out at different pHs, and in order to optimize the NWs composition, solutions with different concentrations of copper were employed. In Table 1, the results obtained in terms of deposition composition and Cu/(Zn + Sn) ratio are reported under different operating conditions (pH and copper concentration). The composition of the NWs was evaluated by EDS. A typical EDS spectrum in which the presence of the characteristic peaks of all the constituent elements of the NWs can be observed is given in Figure 3. Carbon and oxygen peaks originated from the residual polycarbonate that remained on the surface of the electrode after dissolution.

Several studies demonstrated that semiconductors based on CZTS and CZTSe have useful properties as absorbers even at nonstoichiometric composition. In particular, when the deposition processes were carried out using a solution at pH = 3 and with Cu = 1 mM, the values of the Cu/(Zn + Sn) and Zn/Sn ratios indicated that Cu-poor and Zn-rich nanowires were obtained. Such ratios were within the desired ranges (Cu/(Zn + Sn) = 0.7–0.9 and Zn/Sn = 1.2–1.3) for obtaining highly efficient CZTS solar cells [18]. The high efficiency obtained in these conditions can be attributed to the low presence of secondary phases, such as Cu_x_Se and Cu_2_SnSe_3_, which can easily damage the absorber layer. The samples obtained at a pH = 2.26 and Cu = 2 mM (Table 1) showed an excessive copper content, and therefore their Cu/(Zn + Sn) and Zn/Sn ratios were substantially different from the optimal values. The electrodeposition of NWs too rich in copper and too poor in zinc was consistent with the higher thermodynamic redox potential of copper (which favors its cathodic deposition even in the presence of the complexing agent) compared to zinc and with higher concentrations of Cu^2+^ ions in solution. The concentration of tin was in between copper and zinc, in agreement with its electrochemical redox potential. However, it must be highlighted that lactic acid was used as complexing agent to stabilize the solution, which was essential to maintain the Sn^2+^ ion in solution, since, when starting at pH values close to 2, Sn^2+^ ions tend to precipitate as Sn(OH)_2_.

Using a solution at pH = 2.26 with a copper content of 1 mM, the samples obtained were poor in copper and rich in zinc, and the Cu/(Zn + Sn) ratios were closer to the optimal value of 0.7–0.8, while the Zn/Sn ratio was not suitable. The best results were thus obtained working at pH = 3 and Cu =1 mM. From these results, we can conclude that in order to obtain CZTSe NWs with suitable composition, the employment of an electrochemical bath with low content of copper is required.

The SEM images of optimized nanostructured electrodes after the deposition phase are given in Figure 4. Concerning the morphology of the nanostructures, besides having a chemical composition as close as possible to the optimum, the samples must have sufficiently long, intact, and mechanically stable nanostructures. The micrographs show that pulsed current electrodeposition allowed us to obtain mechanically stable nanostructures correctly attached to the Ni support. The Ni surface was thus uniformly covered with nanowires. Due to the addition of sodium sulfate in the electrochemical bath, the nanostructures consisted of only nanowires; in fact, as reported in [8,35,36], sodium sulfate increased the ionic conductivity of the solution and promoted the growth of nanowires rather than nanotubes. The NWs had a cylindrical shape with diameters of about 230 nm, corresponding to the average diameter of the templates. The surface of the NWs reported appeared very smooth, and the typical interconnections of the template pores were clearly visible, as shown in Figure 4. The deposition time ensured the formation of sufficiently long nanostructures (approx. 3 µm) for a good absorption of incident light.

Before depositing the ZnS layer, to avoid a short circuit between the ZnS buffer layer and the Mo back-contact, a CZTSe film was deposited among the NWs and on the surface of the current collector. The film deposition was performed by potentiostatic electrodeposition by applying a potential of −0.65 V (SCE) for 15 min and using the same bath used for the deposition of the nanostructures. Before the deposition and immediately after the dissolution of the polycarbonate template, in order to ensure the wettability of the nanostructures (which was very poor, as expected, considering the morphology of the nanostructures) the electrode was pretreated with a 0.1 M SDBS solution. This step was important because, due to the low wettability, the deposition of the film would take place only on the head of the nanowires with the formation of an compact film that could hinder the access of the electrolyte to the bottom of the nanostructures. Figure 5 shows a CZTSe film growth curve with a shape typical of the deposition of a semiconductor material with poor electronic conductivity [37]. Figure 6 shows the SEM images of NWs after deposition of the CZTSe film. This was uniformly distributed over the entire external surface of the NWs, which appear very rough and thickened, without occluding the interstices between them. The uniform distribution, both on the heads and walls of the nanowires, confirmed the importance of pretreatment with the SDBS solution.

In addition to the SEM and EDS characterization, the samples were also characterized by Raman spectroscopy. Figure 7 reports the Raman spectra of the samples obtained at different pHs and Cu content. The peaks at approx. 280 and 525 cm^−1^ present in samples obtained at pH = 2.26 and Cu = 2 mM were attributable to the secondary phases of CuSe_2_ and Cu_3_Se_2_. Furthermore, it is important to highlight the absence of the characteristic peak of CZTSe. The Raman spectra of the samples deposited at pH = 2.26 and Cu = 1 mM showed the characteristic peak of CZTSe and those of the secondary phases of ZnSe and Cu_3_Se_2_. The spectra of the samples deposited at pH = 3 and Cu = 1 mM showed a more evident CZTSe peak and the disappearance of the ZnSe phase. The peaks corresponding to the Cu_3_Se_2_ were less pronounced, and a new peak was present at 143 cm^−1^, which was attributable to the CuSe_2_ phase. Table 2 summarizes the peak values and the corresponding phases.

The analysis of these spectra showed that the electrodeposition process did not allow us to obtain the pure CZTSe phase, therefore the NWs always presented secondary phases mainly of copper and selenium. This is confirmed by literature, and according to the deposition mechanism proposed by Kroger [38], it could be related to the highly favored electrochemical deposition of metal selenides compared to the deposition of the pure metal.

The annealing process emerged as a fundamental step to modify the crystalline structure and composition of NWs, and had a substantial influence not only on the behavior of the material, but also on the overall features of the final device. Several studies also highlighted that during the annealing process, if the semiconductor was subjected to a sulfurization process, both the crystallinity of the material and photovoltaic performance increased. For example, at IBM, a CZTSSe cell was obtained with an efficiency of 9.6% based on a crystalline film of Cu_2_ZnSn(S,Se)_4_, with an S/(S + Se) = 0.4 ratio, obtained after annealing at 540 °C in the presence of sulfur vapor.

In this research, the sulfurization was carried out in a tubular furnace in an atmosphere of sulfur and argon. In particular, the nanostructured samples were placed in intimate contact with sulfur powder in a 15 cm^3^ inert box. The amount of sulfur powder was carefully calculated in order to reach the desired composition, since sulfur partial pressure must be accurately controlled during the sulfurization process to avoid the evaporation of the volatile SnS, and its negative effect on the reaction of kesterite formation [39,40]. For improving the sulfurization rate, a gas overpressure slightly higher than 1 bar was maintained inside the tubular furnace, while the temperature was kept constant at 540 °C for 40 min. After sulfurization, the samples were slowly cooled down in the furnace.

Table 3 shows the NWs’ composition evaluated by EDS before and after annealing. The comparison shows that after the annealing process, the content of Cu, Zn, and Sn decreased. This was due to the high content of sulfur found in the deposits. In addition, there also was a significant reduction of selenium in the samples, due to its high volatility. The Cu/(Zn + Sn) and Zn/Sn ratios indicated that Cu-poor and Zn-rich NWs were obtained, with very close values to the desired ranges (Cu/(Zn + Sn) = 0.8–0.9, Zn/Sn = 1.2–1.3) to obtain highly efficient CZTSSe solar cells [39].

Figure 8 shows the SEM images of the sample with and without the soft annealing under a sulfur atmosphere. There was no appreciable change in the morphology of the samples that were composed of NWs that were mechanically stable, properly attached to the substrate, and exempt from damages. However, it could be observed that they had partially merged, due to a kesterite formation reaction, and this obviously led to a decrease in the specific area.

The Raman spectra of the samples before and after annealing are reported in Figure 9. The figure shows that the annealing process did not erase the CuSe_2_ and Cu_3_Se_2_ phases completely. Instead, the figure shows the disappearance of the peak at 194 cm^−1^ of the pure CZTSe phase and the appearance of a peak at about 252 cm^−1^, not present in the samples without annealing. This behavior was probably due to the formation of the CZTSSe phase. The presence of sulfur, incorporated during the annealing process, caused a reticular distortion that shifted the peak toward larger values compared to those typical of the pure CZTS phase [41].

For a more precise characterization of the sulfurized NWS, a deconvolution of the Raman spectrum with Lorentzian curves after baseline subtraction and noise filtration was conducted in the region of 100–300 cm^−1^. The deconvolution of the Raman spectrum of samples without and with annealing are shown in Figure 9b,c, respectively. The main peaks of CZTSe without annealing were located at 172, 194, and 252 cm^−1^. This finding strongly supported that the CZTSe structure was the dominant phase before annealing. However, in the deconvoluted curves, we observed that before annealing, the presence of the SnSe phase related to peaks at 122 cm^−1^ was detected, but after annealing, the most intense peaks at 122, 134 and 158 cm^−1^ corresponding to the same compound appeared. After annealing, the main peaks of CZTSe (172 cm^−1^, 194 cm^−1^) were not detected, whereas the peak at 252 cm^−1^ was less intense. This was probably due to the disappearance of the peak related to CZTSe, which was close to that reported for ZnSe still present after annealing.

### 3.2. CIGS Nanowires

A similar procedure used to obtain the NWs of CZTSe was also pursued for the deposition of CIGS nanostructures. In particular, nanostructures were obtained on a Mo-modified nickel current collector by pulsed current deposition (from 0 to −1.53 mA/cm^2^). The electrodeposition was carried out in a solution at pH = 2.7 with increasing concentrations of gallium. Table 4 shows the results (sample composition, Cu/(In + Ga) ratio and Ga/(In + Ga) ratio) obtained with different gallium concentrations.

CIGS-based materials have useful properties as absorbers even at nonstoichiometric compositions. In particular, a key role in such regard is attributed to the content of copper and gallium. In fact, the highest efficiency (above 20%) of CIGS devices was obtained with Ga/(In + Ga) and Cu/(In + Ga) ratios of about 0.3 and 0.7, respectively (Eg = 1.1–1.25 eV). The good performances were attributed to the low presence of the secondary phases [42,43]. In our case, Table 4 shows that all samples had an excessive copper content with a consequent Cu/(In + Ga) ratio much greater than 0.7. This result was consistent with the high electrochemical redox potential of copper compared to indium and gallium. To obtain CIGS deposits with composition useful as absorber material, it was necessary to reduce the copper concentration in solution and optimize the amount of gallium to maintain a Cu/(In + Ga) ratio of about 0.3. The composition reported in Table 4 was calculated by means of EDS analysis (Figure 10).

Figure 11 shows the SEM images of the as-prepared CIGS nanostructures. The pulsed current electrodeposition allowed us to obtain mechanically stable nanostructures, appropriately fixed to the substrate and uniformly distributed over the entire surface of the current collector. Nanostructures consisted of NWs with a morphology that reflected the shape of the template, therefore showing the characteristic interconnections and an average diameter size of about 230 nm. The height of the nanostructures was about 3 μm and was uniformly distributed over the entire surface of the sample.

Figure 12 shows the Raman spectrum of the NWs obtained at pH = 2.7 with a gallium content of 57 mM and at 1.53 mA/cm^2^. The characteristic peak of CIGS, at about 173 cm^−1^, can be observed even with a low intensity, probably due to the low crystallinity of the deposit. The other peaks present were attributable to the secondary phases of CuSe_2_, CuSe, or CuSe_2_ and Cu_3_Se_2_. The presence of these phases was obviously due to the high content of copper and selenium present in the NWs, and was consistent with the literature data. A photoelectrochemical behavior of similar nanostructures showed a cathodic photocurrent and an optical gap of 1.55 eV [43,44,45].

The deconvolution of the Raman spectrum of the sample is shown in Figure 12b. The spectra showed an A1 mode at approximately 174 cm^−1^, generally observed in the I-II-VI chalcopyrite compounds [46]. The peak at 220 cm^−1^ corresponded to the B2 mode of the CuInSe_2_ phase, as reported in [47]. The elementary selenium was identified by the peak at 236 cm^−1^, which was due to trigonal selenium [48]. Additional peaks at 145 and 258 cm^−1^ could be related to the presence of the Cu_x_Se secondary phase, with a symmetry of lattice vibrations different from chalcopyrite [49].

Similar to the CZTSe nanostructures, after the dissolution of the membrane, a thin film of CIGS was deposited on the surface of the nanostructures and on the base of the current collector to which they were attached. This layer was essential to avoid the contact between the modified current collector with Mo, which acted as a back-contact, and the n part of the junction (ZnS). Before the deposition of the CICG film, the nanostructures were treated with SDBs in order to increase their wettability. To increase the crystallinity of the deposit and reduce the presence of secondary phases, a heat treatment was also performed in an inert atmosphere at 540 °C for 40 min. Afterward, the ZnS buffer layer was deposited by CBD, and the morphology of the nanowires did not appear significantly modified by the buffer layer deposition, as shown in Figure 13. In particular, the presence of a deposit above the surface of the nanostructures could be observed. In addition, the covering of nanowires with an In_2_S_3_ buffer layer was under investigation. In particular, the possibility to obtain a thin and uniform layer of In_2_S_3_ by ion layer gas reaction on Ni nanowires has been successful demonstrated [50,51]. This could be a good solution, because as demonstrated, the deposition of In_2_S_3_ on top of Cu(In,Ga)(S,Se)_2_ absorber layers resulted in solar cells with a certified efficiency of 16.1% [52].

### 3.3. CZTSe/ZnS/ZnO-AZO Fabrication

To complete the nanowire-based cell structure, ZnO and ZnO:Al (AZO) films were deposited using RF magnetron-sputtering. This last step was done only on the CZTSe nanostructures, because they had an almost optimal composition, while the composition of the CIGS nanostructures had not been optimized yet. The ZnO thin film was obtained at room temperature using 150 W (in RF), with a deposition time of 60 min at a pressure of 5 mTorr. These parameters allowed us to obtain a 100 nm-thick ZnO film. The AZO was deposited using a power of 140 W (in RF), deposition times of 40–75 min, and at a pressure of 2.4 mTorr, and maintaining the substrate temperature at 225 °C. With such parameters, a thickness in the range of 500–750 nm was obtained. The CZTSe NWs covered with AZO film were characterized from a morphological and compositional point of view. The comparison of the EDS spectra of the NWs pre- and postdeposition of the AZO sample showed the presence of only one new peak, which could be attributed to the presence of aluminum.

Figure 14 shows the SEM images of the sample after the AZO deposition, in which no appreciable change in the morphology of the deposit can be seen. The AZO film was uniformly distributed over the entire outer surface of the nanostructures, and appeared very rough.

As expected, the AZO film did not significantly modify the structure and morphology of the sample due to its small thickness (hundreds of nm) compared to the thickness of the nanostructures (a few µm). The future development of this work will be the deposition of the front contact and the electrical characterization of the cells obtained here in order to obtain the I–V characteristic.

## 4. Conclusions

CZTSe and CIGS nanowires were fabricated on Mo-modified Ni current collectors by a one-step electrodeposition method, by means of a nanoporous polycarbonate membrane as a template. Compared with other deposition methods, the electrodeposition in the PC template was simple, low-cost, and versatile. The effect of precursor concentration in the electrolyte and the pH on the composition of the nanostructures was carefully investigated. Deposited NWs were annealed at 540 °C (under a sulfur atmosphere in the case of CZTSe and in an inert atmosphere for CIGS) to increase their crystallization and to reduce the presence of secondary phases, mainly composed of copper and selenium. The structural, morphological, and compositional properties of the NWs before and after each deposition were examined by SEM, EDS, and Raman spectroscopy. The as-prepared NWs presented a perfectly cylindrical morphology with a uniform diameter throughout their length. The composition and crystalline structures of NWs changed after the thermal treatment, while the annealing process did not alter their morphology. Raman analysis revealed the presence of Cu_2−x_Se phases, also after annealing. The ZnS(O,OH) was deposited by chemical bath at 95 °C for 90 min. Finally, ZnO and ZnO:Al films were deposited through RF magnetron-sputtering. Our approach suggests the possibility of production of nanowire solar cells via electrodeposition that could provide high-quality CZTSe and CIGS with low capital investment due to the use of low-cost starting materials, a low deposition temperature, and minimal waste generation (the solution can be recycled).

## Figures and Tables

**Figure 1 materials-14-02778-f001:**
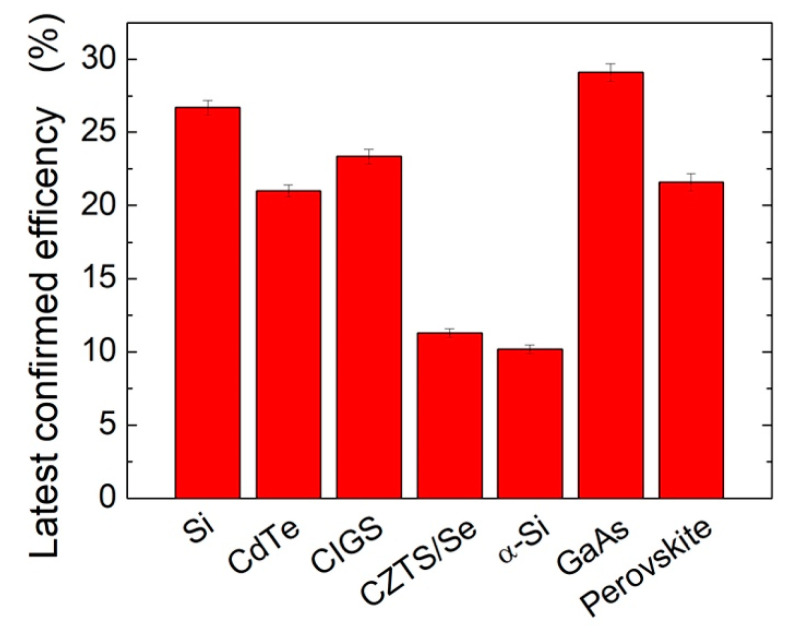
Efficiency of different materials in single-junction solar cells.

**Figure 2 materials-14-02778-f002:**
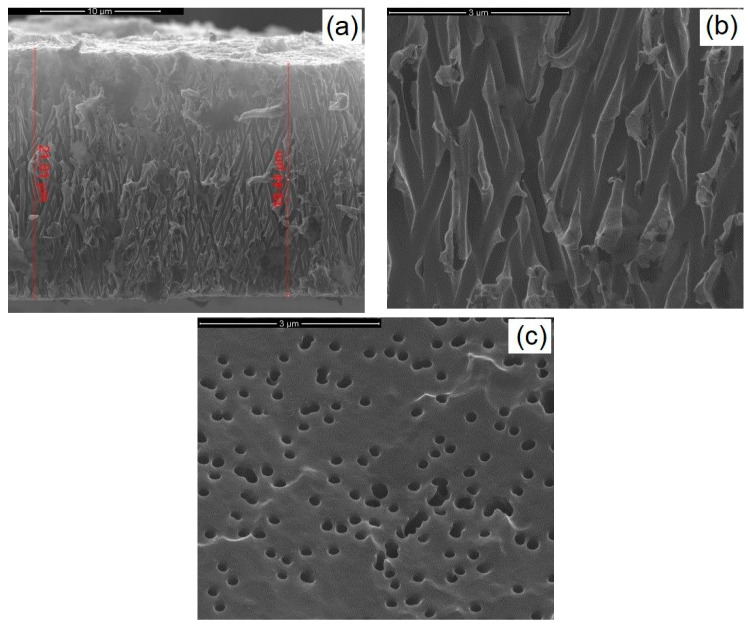
Micrographs of a polycarbonate membrane (**a**) with cross-section (**b**) and top (**c**) views.

**Figure 3 materials-14-02778-f003:**
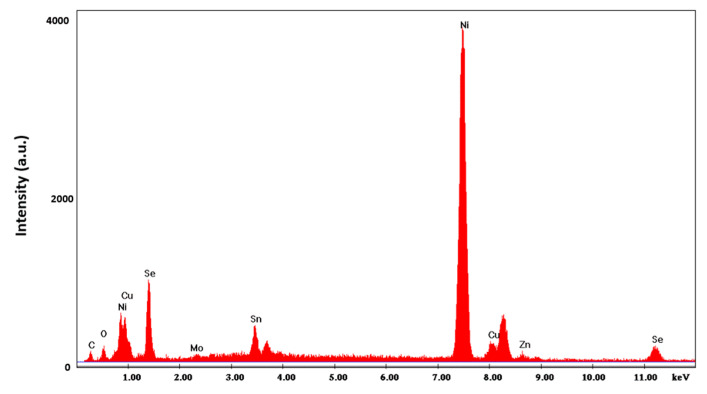
EDS spectrum of the optimized CZTSe nanostructured electrode. NWs were obtained at pH = 3 with a copper content of 1 mM.

**Figure 4 materials-14-02778-f004:**
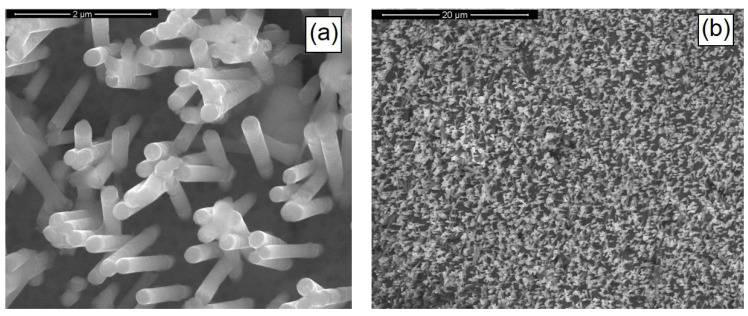
SEM images of CZTSe NWs. (**a**) High and (**b**) low magnification. The NWs were obtained at pH = 3 with a copper content of 1 mM.

**Figure 5 materials-14-02778-f005:**
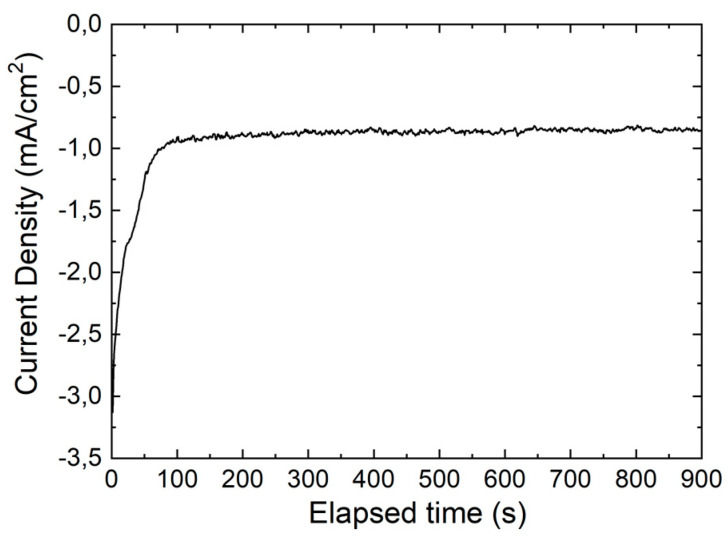
CZTSe film growth curve. The deposition was carried out applying a potential of −0.65 V (SCE) for 15 min.

**Figure 6 materials-14-02778-f006:**
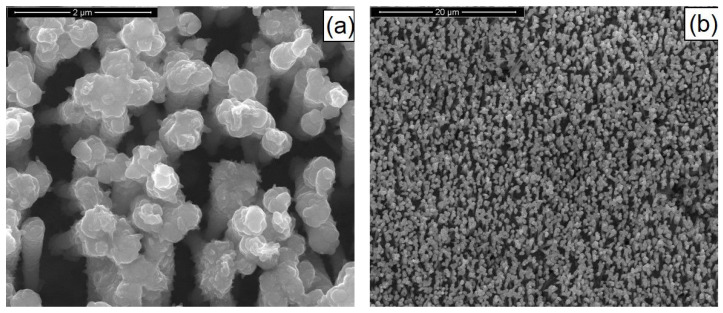
SEM images of CZTSe NWs sample after CZTSe film deposition. High (**a**) and low (**b**) magnification. NWs were obtained at pH = 3 with a copper content of 1 mM.

**Figure 7 materials-14-02778-f007:**
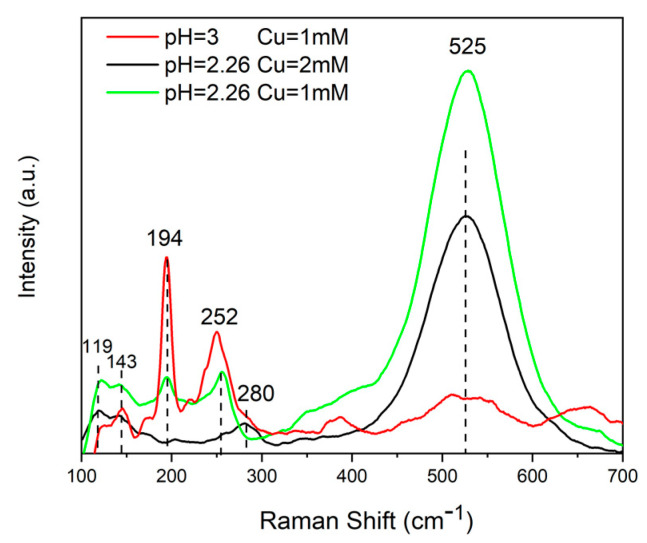
Raman spectra of the CZTSe NWs obtained using an electrodeposition bath at different pHs and Cu content (see Table 1).

**Figure 8 materials-14-02778-f008:**
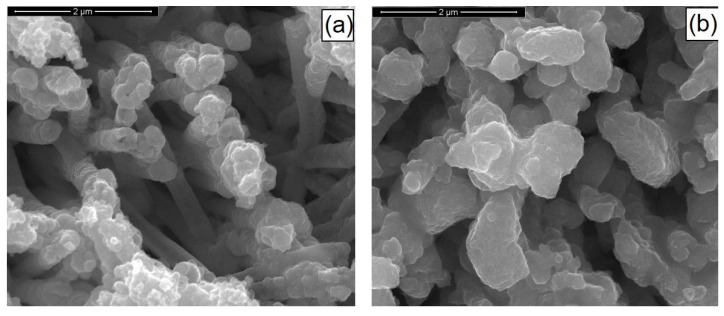
SEM images of CZTSe nanostructures (**a**) before and (**b**) after annealing. NWs were obtained at pH = 3 with a copper content of 1 mM.

**Figure 9 materials-14-02778-f009:**
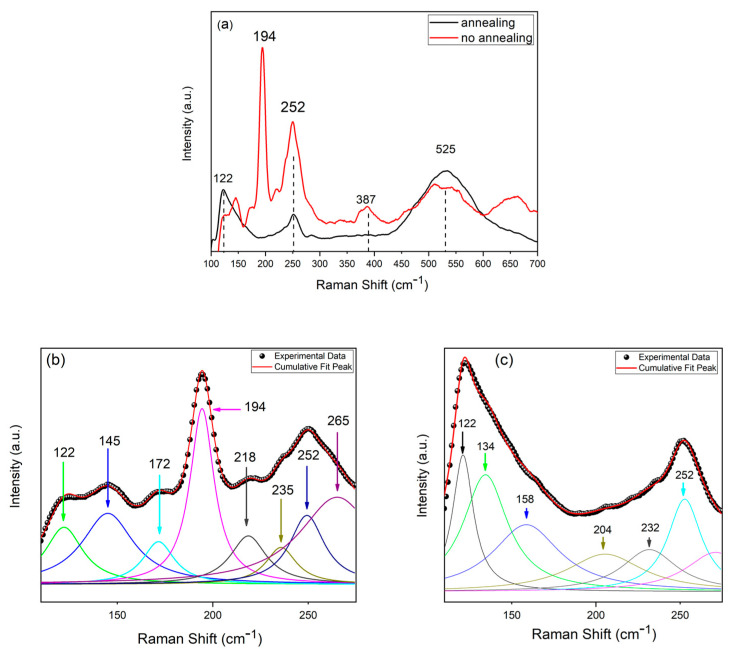
Raman spectrum before and after annealing of the nanowires: (**a**) original spectra, deconvolution of the Raman spectrum of (**b**) no-annealed sample and (**c**) annealed sample. NWs were obtained at pH = 3 with a copper content of 1 mM.

**Figure 10 materials-14-02778-f010:**
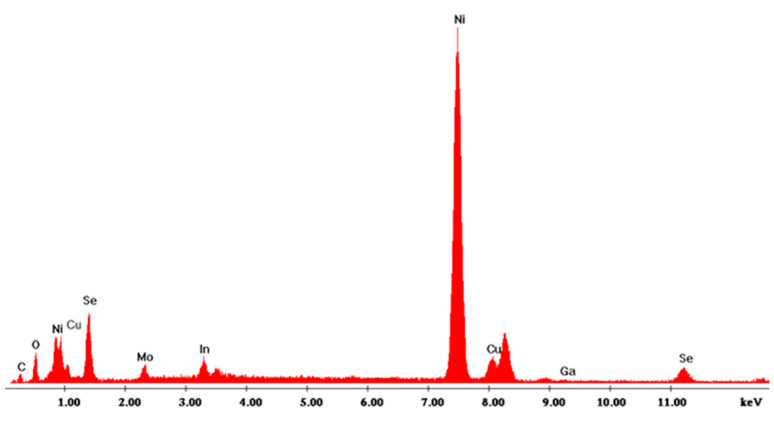
EDS spectrum of CIGS NWs. NWs were obtained at pH = 2.7 with a gallium content of 57 mM and at 1.53 mA/cm^2^.

**Figure 11 materials-14-02778-f011:**
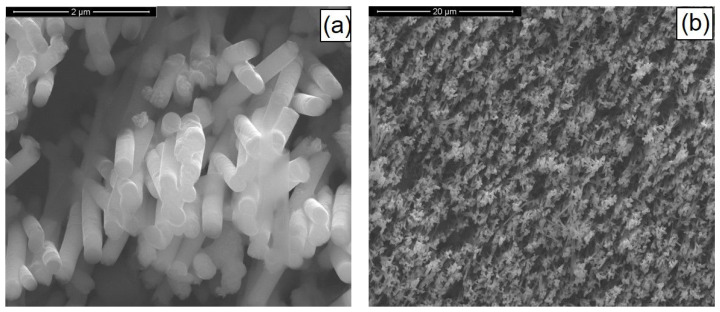
SEM images at high (**a**) and low (**b**) magnifications of CIGS-based nanostructures. NWs were obtained at pH = 2.7 with a gallium content of 57 mM and at 1.53 mA/cm^2^.

**Figure 12 materials-14-02778-f012:**
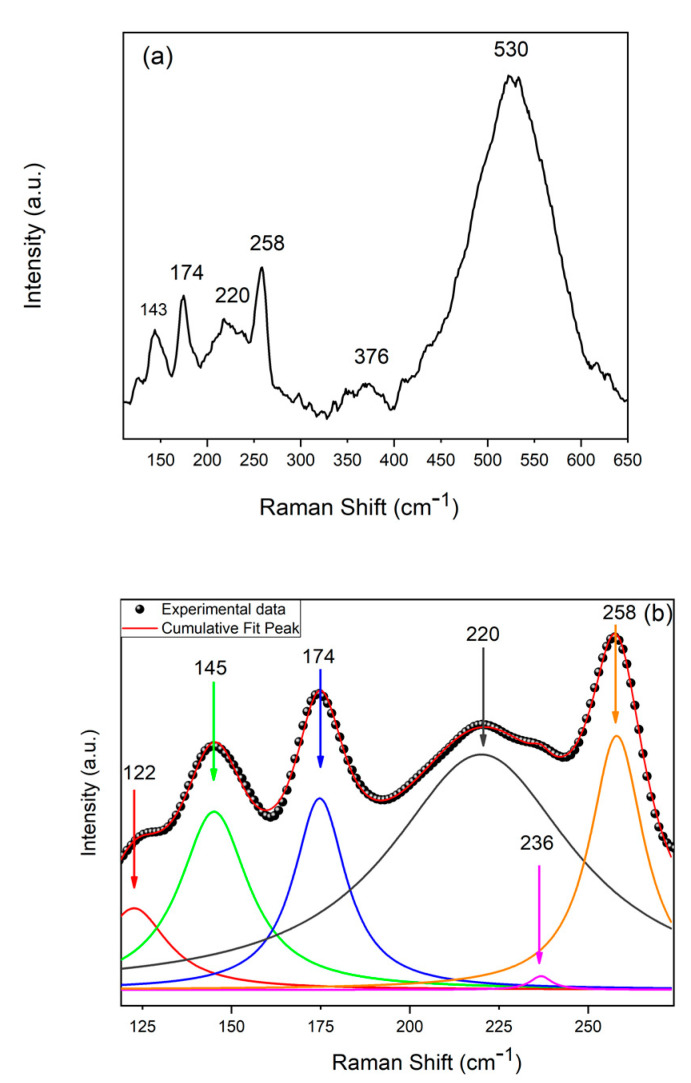
Raman spectra of samples: (**a**) original spectrum and (**b**) deconvolution. NWs were obtained at pH = 2.7 with a gallium content of 57 mM and at 1.53 mA/cm^2^.

**Figure 13 materials-14-02778-f013:**
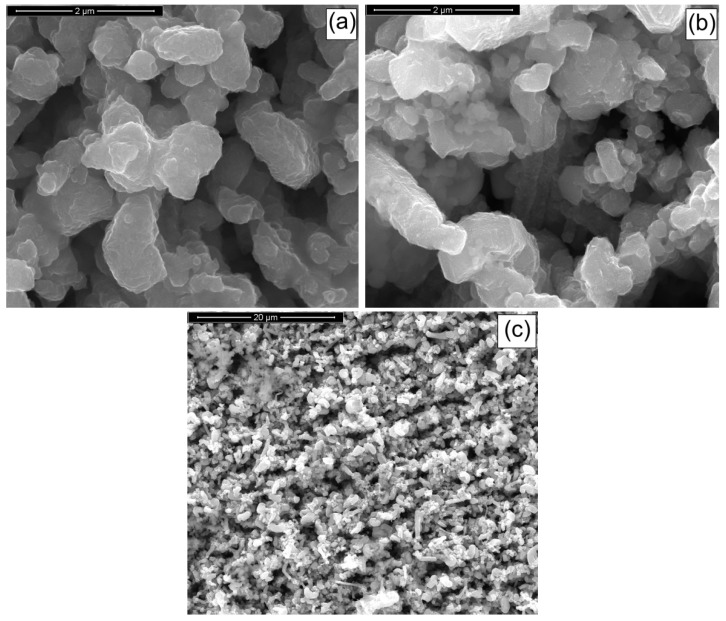
SEM images of the NWs (**a**) after the deposition of CIGS thin film (**b**) and after the ZnS deposition. (**c**) Low-magnification SEM images of the sample before the ZnS film. NWs were obtained at pH = 2.7 with a gallium content of 57 mM and at 1.53 mA/cm^2^.

**Figure 14 materials-14-02778-f014:**
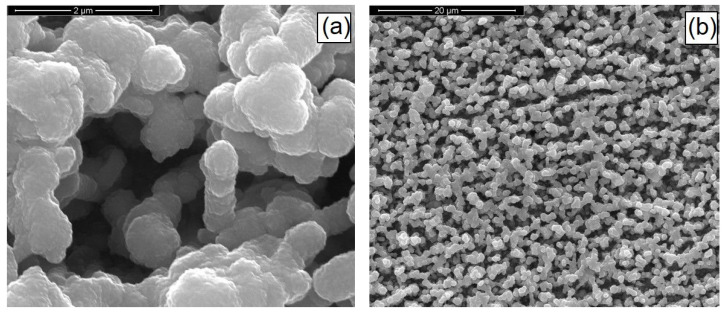
(**a**) High- and (**b**) low-magnification SEM images after the deposition of ZnO/AZO film. CZTSe NWs were obtained at pH = 3 with a copper content of 1 mM.

**Table 1 materials-14-02778-t001:** Composition of as-deposited CZTSe NWs obtained in different electrochemical conditions.

pH	Cu (mM)	Cu (%)	Zn (%)	Sn (%)	Se (%)	Cu/(Zn+Sn)	Zn/Sn
2.26	2	53.2 ± 1.6	1.5 ± 0.5	29.1 ± 1.3	16.2 ± 1.2	1.74 ± 0.6	0.05 ± 0.03
2.26	1	36.2 ± 1.2	9.1 ± 0.9	11.5 ± 1.2	43.2 ± 1.6	1.75 ± 0.5	0.8 ± 0.1
3	1	26.8 ± 1.5	15.9 ± 1.1	12.4 ± 1.6	44.9 ± 1.6	0.94 ± 0.3	1.28 ± 0.5

**Table 2 materials-14-02778-t002:** Summary of Raman analysis of CZTSe nanowires.

Compound	Raman Shift (cm^−1^)
**CZTSe**	172, 194–197, 231–235, 239–254
**CuSe**	240–260
**CuSe_2_**	142, 233, 260–270, 450
**Cu_2-x_Se**	200, 260, 460
**Cu_3_Se_2_**	190, 510, 560
**Cu_2_SnSe_3_**	180, 200–230, 250, 360
**Cu_x_Se**	260
**ZnSe**	252–256
**SnSe**	130, 150
**SnSe_2_**	119, 185
**Se**	235, 253, 440

**Table 3 materials-14-02778-t003:** Cu/(Zn + Sn) and S/(S + Se) ratios before and after annealing. NWs were obtained at pH = 3 with a copper content of 1 mM.

Cu/(Zn + Sn) Before	Cu/(Zn + Sn) After	Zn/Sn Before	Zn/Sn After	S/(S + Se)
**0.94 ± 0.06**	0.22 ± 0.03	1.28 ± 0.07	0.9 ± 0.06	0.47 ± 0.06

**Table 4 materials-14-02778-t004:** EDS analysis of CIGS NWs obtained in different conditions.

Ga (mM)	Cu (%)	In (%)	Ga (%)	Se (%)	Cu/(In + Ga)	Ga/(In + Ga)
**5.7**	37.2 ± 1.4	12.2 ± 0.9	0.7 ± 0.2	49.6 ± 1.4	2.9 ± 0.8	0.054 ± 0.03
**57**	34.7 ± 1.2	12.6 ± 0.9	3.3 ± 0.3	49.4 ± 1.8	2.2 ± 0.6	0.2 ± 0.08
**114**	38.4 ± 1.5	9.8 ± 0.8	1 ± 0.3	50.8 ± 1.6	3.5 ± 0.8	0.09 ± 0.03

## Data Availability

Data is contained within the article.

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
