# Peer review of "Fabrication of CZTSe/CIGS Nanowire Arrays by One-Step Electrodeposition for Solar-Cell Application"

_materials, 2021, doi:10.3390/ma14112778_

Round 1
Reviewer 1 Report
Oliveri et al. in the paper draft "Fabrication ofCZTSe/CIGS Nanowire Arrays by One Step Electrodeposition for Solar Cells Application" presented
some important results and particularly, one step electrodeposition technique.
After going through the manuscript, suggest a major revision to enrich the manuscript.
Statement: "The value of its bandgap (1.1 eV) is close to the optimum value for a single photovoltaic absorber[1]." is not true. Higher bandgap (~1.5 - 1.6 are good for efficient absorption).
In the third part of the introduction provide the latest efficiency of CIGS. Also, similar materials CdTe, C-Si, a-Si:H and the latest efficiency of CZTSe/CZTS should be stated here with reference.
Images in Fig. 1 or in others, please isolate them. (they overlapped)
Fig. 8: Raman spectra need further analysis. Whet after annealing, the intensity above ~ 220 drops. Convolution spectra?
For kesterite, the band-fluctuation is one of the major losses. The author needs to provide some discussion here. The electrical estimation can be found in the article ( https://doi.org/10.1016/j.tsf.2019.01.052 ). The author should consider their data in table 5 and discuss.
Fig.11 caption: provide details about it even though in the text. Suggest doing for other figures as well.
Finally, preliminary JV data for PV performance need to add.
Author Response
Oliveri et al. in the paper draft "Fabrication ofCZTSe/CIGS Nanowire Arrays by One Step Electrodeposition for Solar Cells Application" presented some important results and particularly, one step electrodeposition technique.
After going through the manuscript, suggest a major revision to enrich the manuscript.
Response: Many thanks to reviewer for his positive feedback and for all his suggestions that allowed us to improve our article.
Statement: "The value of its bandgap (1.1 eV) is close to the optimum value for a single photovoltaic absorber[1]." is not true. Higher bandgap (~1.5 - 1.6 are good for efficient absorption).
Response: In the revised text this statement was cancelled.
In the third part of the introduction provide the latest efficiency of CIGS. Also, similar materials CdTe, C-Si, a-Si:H and the latest efficiency of CZTSe/CZTS should be stated here with reference.
Response: In the revised text we have inserted a new Figure (Figure 1) reporting the latest efficiency of the solar cell made with some selected materials.
Images in Fig. 1 or in others, please isolate them. (they overlapped)
Response: In the revised text all figures have been separated
Fig. 8: Raman spectra need further analysis. Whet after annealing, the intensity above ~ 220 drops. Convolution spectra?
Response: In the revised text, we have modified Figure 9 (Figure 8 in the original text). Besides the convolution of the Raman peaks were performed.
For kesterite, the band-fluctuation is one of the major losses. The author needs to provide some discussion here. The electrical estimation can be found in the article (https://doi.org/10.1016/j.tsf.2019.01.052 ). The author should consider their data in table 5 and discuss.
Response: We have not studied this aspect, because it is beyond the scope of this work. The aim of this work is to demonstrate that starting from the very simple method of electrochemical deposition it is possible to obtain regular arrays of CIGS and CZTS nanowires with controlled composition, that have been also modified with successive layer in order to obtain the p-n junction. For this reason, in this work we have reported the results of the accurate chemical-physical characterization of the sample after the different steps of fabrication. The electrical and PV characterization of the samples is under study with the aim to correlate the PV performance with the deposition parameters
Fig.11 caption: provide details about it even though in the text. Suggest doing for other figures as well.
Response: In the revised text in all caption more details were added
Finally, preliminary JV data for PV performance need to add.
Response: The electrical and PV characterization of the samples is under study with the aim to correlate the PV performance with the deposition parameters.
Reviewer 2 Report
The development of nanowire arrays for solar cell applications have attracted considerable attention and promoted encouraging progress in recent years. In this manuscript entitled “Fabrication of CZTSe/CIGS Nanowire Arrays by One Step Electrodeposition for Solar Cells Application”, Rosalinda Inguanta and co-workers proposed the use of one step electrodeposition to prepare CZTSe/CIGS nanowire arrays. The nanowire composition was tuned by adjusting applied potential and electrolyte composition. To control the morphology of the nanostructures, the electrodeposition was carried out by applying rectangular-shaped current pulses. Based upon this, they characterized the nanostructure by a set of advanced techniques such as Scanning Electrode Microscopy (SEM), Raman Spectroscopy, and Energy Dispersion Spectroscopy (EDS). In addition, a solar cell with a Al:ZnO/ZnS/CZTS/Mo structure derived from the CZTS/CIGS synthetized nanowires was demonstrated.
In my opinion, the manuscript is systematic that may help the community to understand and optimize the nanowire arrays for energy applications. Moreover, the paper is technically sound. Thus, it might be suitable for a publication in Materials if the following comments are properly addressed.
Some points to note
- The paper is organized like an experimental report not a scientific paper.
- The standard deviation of each parameter (Table 6) should be added.
- In Table 2, the ratio should be 0.94, 0.22, 1.28, NOT 0,94, 0,22, 1,28.
Author Response
The development of nanowire arrays for solar cell applications have attracted considerable attention and promoted encouraging progress in recent years. In this manuscript entitled “Fabrication of CZTSe/CIGS Nanowire Arrays by One Step Electrodeposition for Solar Cells Application”, Rosalinda Inguanta and co-workers proposed the use of one step electrodeposition to prepare CZTSe/CIGS nanowire arrays. The nanowire composition was tuned by adjusting applied potential and electrolyte composition. To control the morphology of the nanostructures, the electrodeposition was carried out by applying rectangular-shaped current pulses. Based upon this, they characterized the nanostructure by a set of advanced techniques such as Scanning Electrode Microscopy (SEM), Raman Spectroscopy, and Energy Dispersion Spectroscopy (EDS). In addition, a solar cell with a Al:ZnO/ZnS/CZTS/Mo structure derived from the CZTS/CIGS synthetized nanowires was demonstrated.
In my opinion, the manuscript is systematic that may help the community to understand and optimize the nanowire arrays for energy applications. Moreover, the paper is technically sound. Thus, it might be suitable for a publication in Materials if the following comments are properly addressed.
Response: Many thanks to reviewer for his positive feedback and for all his suggestions that allowed us to improve our article.
Some points to note
- The paper is organized like an experimental report not a scientific paper.
Response: In this paper very different experiments were carried out both for the deposition of the nanostructures and for their modification with the successive layer to obtain the p-n junction. Considering the very high quantity of the data, we have separately discussed each step of fabrication also because each stage has required a specific optimization process
- The standard deviation of each parameter (Table 6) should be added.
Response: In the revised text the standard deviation of each parameter was reported
- In Table 2, the ratio should be 0.94, 0.22, 1.28, NOT 0,94, 0,22, 1,28.
Response: In the revised text, we have corrected these mistakes

Reviewer 3 Report
A novel method to fabricate the nanowire part of the CIGS/CZTS/Se thin-film solar cells is discussed in the manuscript. The nanowire array was characterized by SEM, EDX, and Raman measurements but no device characterization of these nanowires was provided. However, a major revision needs to be undertaken before this manuscript can be considered for publication.
- Please use short and simple sentences to explain different figures and insights. There are many grammatical errors, and it is advised to get help from the English editing service associated with MDPI Materials.
- Please use consistent numerical notations throughout the manuscript. In some sentences, the decimals are indicated by a point and in others by a comma.
- The dimensions of the nanowire are big enough to enable higher-quality SEM images with better clarity. The SEM images in the manuscript show a great amount of blurring, directional stretching, and beam misalignment. If possible, please repeat the SEM of these nanowires with the required beam alignment, lens alignment, and astigmatism. This is especially required for figure 3a and also some of the other SEM images.
Author Response
A novel method to fabricate the nanowire part of the CIGS/CZTS/Se thin-film solar cells is discussed in the manuscript. The nanowire array was characterized by SEM, EDX, and Raman measurements but no device characterization of these nanowires was provided. However, a major revision needs to be undertaken before this manuscript can be considered for publication.
- Please use short and simple sentences to explain different figures and insights. There are many grammatical errors, and it is advised to get help from the English editing service associated with MDPI Materials.
Response: In the revised text the English was careful checked.
- Please use consistent numerical notations throughout the manuscript. In some sentences, the decimals are indicated by a point and in others by a comma.
Response: In the revised text, we have corrected these mistakes
- The dimensions of the nanowire are big enough to enable higher-quality SEM images with better clarity. The SEM images in the manuscript show a great amount of blurring, directional stretching, and beam misalignment. If possible, please repeat the SEM of these nanowires with the required beam alignment, lens alignment, and astigmatism. This is especially required for figure 3a and also some of the other SEM images.
Response: In the revised text, the SEM images were replaced with higher-quality format

Round 2
Reviewer 1 Report
Agreed!
Reviewer 2 Report
The authors have fully addressed my concerns in the first round. Thus, I am very happy to recommend it for publication in the current form. No further changes are required.
Reviewer 3 Report
The authors have made edits requested and SEM images looks better. However, it is hard to evaluate the manuscript in the current format retaining strike through words/sentences/paragraphs, images that were newly added and figures included from the previous version. Please resubmit the manuscript after removing old images, strike through words/sentences/paragraphs to enable readability and clarity.